# Adrenal Venous Sampling Could Be Omitted before Surgery in Patients with Conn’s Adenoma Confirmed by Computed Tomography and Higher Normal Aldosterone Concentration after Saline Infusion Test

**DOI:** 10.3390/diagnostics12071718

**Published:** 2022-07-15

**Authors:** Robert Holaj, Petr Waldauf, Dan Wichterle, Jan Kvasnička, Tomáš Zelinka, Ondřej Petrák, Zuzana Krátká, Lubomíra Forejtová, Jan Kaván, Jiří Widimský

**Affiliations:** 1Centre for Hypertension, 3rd Department of Medicine, General University Hospital and 1st Faculty of Medicine, Charles University, U Nemocnice 504/1, 128 08 Prague, Czech Republic; jan.kvasnicka3@vfn.cz (J.K.); tzeli@lf1.cuni.cz (T.Z.); ondrej.petrak@vfn.cz (O.P.); zuzana.kratka@vfn.cz (Z.K.); jwidi@lf1.cuni.cz (J.W.J.); 2Department of Anesthesiology, University Hospital Královské Vinohrady and 3rd Faculty of Medicine, Charles University, Šrobárova 1150/50, 100 00 Prague, Czech Republic; petrwaldauf@gmail.cz; 3Department of Cardiology, Institute for Clinical and Experimental Medicine, Vídeňská 1958/9, 140 21 Prague, Czech Republic; dan.wichterle@ikem.cz; 4Department of Radiology, General University Hospital and 1st Faculty of Medicine, Charles University, U Nemocnice 499/2, 128 08 Prague, Czech Republic; lubomira.forejtova@vfn.cz (L.F.); jan.kavan@vfn.cz (J.K.)

**Keywords:** adrenal venous sampling, aldosterone-producing adenoma, idiopathic aldosteronism, prediction score, primary aldosteronism, saline infusion test

## Abstract

Purpose: Adrenal venous sampling (AVS) performed to distinguish unilateral and bilateral primary aldosteronism (PA) is invasive and poorly standardized. This study aimed to identify non-invasive characteristics that can select the patients with unilateral PA who could bypass AVS before surgery. Methods: A single-center study collected a total of 450 patients with PA. Development and validation cohorts included 242 and 208 patients. The AVS was successful in 150 and 138 patients from the cohorts, and the unilateral PA was found in 96 and 94 patients, respectively. Clinical factors independently associated with lateralized AVS in multivariable logistic regression were used to construct a unilateral PA prediction score (SCORE). Results: The proposed SCORE was calculated as a sum of the prevalence of adrenal nodule on computed tomography (2 points) and plasma/serum aldosterone concentration ≥ 165 ng/L after the saline infusion test (SIT) (1 point). Importantly, the SCORE = 3 points identified 48% of unilateral PA patients with a specificity of 100% in the development cohort. The zero rate of false-positive classifications was preserved with the same cut-off value in the validation cohort. Conclusions: AVS could be omitted before surgery in patients with typical Conn´s adenoma provided the aldosterone concentration ≥ 165 ng/L after the SIT.

## 1. Introduction

Primary aldosteronism (PA) is a disease characterized by autonomous aldosterone overproduction leading to moderate, severe, or resistant arterial hypertension [1]. High aldosterone levels also result in increasing potassium losses in urine, which may cause hypokalemia, as well as the activation of an inflammatory system response with collagen deposits in the vascular and myocardial wall [2], leading to vascular and myocardial fibrosis [3,4], vascular and left ventricle remodeling [5,6,7,8], and other pathological phenomena.

From the epidemiological perspective, PA is the most frequent form of secondary hypertension resulting from endocrine causes. Its prevalence in a non-selected population of patients with arterial hypertension is about 6% [9,10,11], but the prevalence in the population of patients with moderate or severe arterial hypertension reaches 20% [12,13,14].

Laboratory diagnostics of PA consist of determining plasma (or serum) aldosterone concentration (PAC) and plasma renin activity (PRA) or direct renin concentration (DRC), and the ratio of PAC to PRA or PAC to DRC (ARR), respectively. Validation of PA by confirmation test should follow in case of ambiguous findings. Insufficient suppression of aldosterone secretion during the saline infusion test (SIT) is considered as a definite confirmation of the PA diagnosis [15].

Aldosterone overproduction in the adrenal gland can be unilateral or bilateral. Unilateral PA is usually caused by a unilateral aldosterone-producing adenoma (APA), while bilateral PA, named idiopathic hyperaldosteronism (IHA), is mostly caused by bilateral hyperplasia of the adrenal glands [16].

The two basic forms of PA differ fundamentally in the therapeutic procedure applied, which can consist of adrenalectomy or the application of mineralocorticoid receptor blockers [15]. In addition, the accuracy of computed tomography (CT) in an indication of laterality of the disease is limited because of the possibility of missing a microadenoma or, on the other hand, discoveringa non-functioning adenoma [17]. Therefore, the best method used for discrimination between the APA and IHA (including laterality information) is still adrenal venous sampling (AVS). This examination is, however, complicated, technically challenging, and expensive [18,19].

Considering the limitations of AVS, it would be most helpful to identify patients with a high certainty of the diagnosis of either unilateral or bilateral PA in whom AVS could be avoided. Nevertheless, the existing clinical prediction scores vary in accuracy according to the databases, and therefore, their clinical applicability is limited [20,21,22,23,24,25,26,27]. In this study, we aimed to develop and validate a clinical prediction score (SCORE) for classifying the PA subtype which would be potentially superior to already published prediction methods.

## 2. Materials and Methods

### 2.1. Patients

We retrospectively analyzed two cohorts of patients investigated for severe or resistant hypertension and suspected secondary hypertension in our tertiary-hospital-based Centre for Hypertension. The development cohort consisted of patients with a suspected diagnosis of PA, based on elevated aldosterone to PRA ratio ≥30 ng/dL/(ng/mL/h), who were investigated between November 2003 and June 2011. The validation cohort consisted of patients with a suspected diagnosis of PA, based on elevated aldosterone to DRC ratio ≥5.7 ng/dL)/(ng/mL), who were investigated between July 2011 and December 2020. PRA or DRC and aldosterone levels were measured after overnight recumbency and a 2-h standing position [28]. In all the patients, any other main form of secondary hypertension (pheochromocytoma, renal parenchymal disease, renovascular hypertension) or drug-induced hypertension was carefully excluded. Each patient underwent a 1 mg dexamethasone suppression test to rule out Cushing syndrome. Each participant signed written informed consent with the study. The original study protocol and the protocol amendment were approved by the local Ethics Committee of the General University Hospital in Prague on 26 June 2003 and 21 June 2012, respectively.

### 2.2. Drugs Management

Antihypertensive medication influencing the levels of examined hormones was discontinued two weeks before admitting the patients to the hospital (6 weeks, in the case of spironolactone treatment). In fertile females, oral contraceptives were interrupted for two months. Patients were switched to treatment by an α-blocker (doxazosine) and/or a slow-release non-dihydropyridine calcium channel blocker (verapamil). In the case of hypokalemia, patients continued oral substitution of potassium [29].

### 2.3. Postural Stimulation Test (PST)

PAC, PRA or DRC, and cortisol were measured at 07:00 AM after all-night bed rest and then again after 2 h of walking. A percentage of PAC increase in response to posture (to quantify the renin-mediated aldosterone responses) was calculated and considered interpretable only if cortisol (as an indirect index of ACTH secretion) followed its normal circadian rhythm and did not increase during the test. PAC increase of ≤30% supports the presence of unilateral PA.

### 2.4. Saline Infusion Test (SIT)

The test was carried out the following day. PAC was measured before and after a rapid administration of 2 L of saline infusion over 4 h. The diagnosis of PA was confirmed when PAC did not drop below 5 ng/dL.

### 2.5. Adrenal Venous Sampling (AVS)

Aldosterone and cortisol concentrations were measured in blood samples from both adrenal veins and the inferior vena cava. AVS examination was carried out without adrenocorticotrophic hormone stimulation. The results were evaluated in line with the current expert consensus [18,30,31], according to which the unilateral aldosterone overproduction is indicated by the lateralization index > 4 calculated as [aldosterone(dominant side)/cortisol(dominant side)]/[aldosterone(non-dominant side)/cortisol(non-dominant side)] together with selectivity index > 2 calculated as cortisol(adrenal vein)/cortisol(inferior vena cava) and assessed for both adrenal veins, if the patients were biochemically cured according to the Primary Aldosteronism Surgical Outcome (PASO) criteria 6–12 months after unilateral adrenalectomy [32]. AVS in which lateralization index was between 2 and 4 was considered as unilateral aldosterone overproduction provided other parameters and clinical data such as the presence of contralateral suppression < 1 (what may indicate unilateral aldosteronism on the opposite side), hypokalemia below 3.0 mmol/L and persistence of uncontrolled hypertension on MRA therapy were considered [31,33].

### 2.6. Laboratory Methods

Until June 2011, all endocrine laboratory examinations (PAC, PRA, and plasma cortisol) were performed by radioimmunological analysis using commercially available kits (Immunotech; Beckman Coulter Company, Prague, Czech Republic) in the dedicated local laboratory. Since July 2011, all endocrine laboratory examinations (DRC, serum aldosterone, and cortisol) were performed by chemiluminescence assay using commercially available kits (DiaSorin; Saluggia, Italy) in our central laboratory.

### 2.7. Blood Pressure Measurement

Casual blood pressure was measured using an oscillometric device (Omron M6, Shimogyo-ku, Kyoto, Japan). The measurement was taken in a noiseless, quiet room with the patient’s arm situated at their heart level. Blood pressure was measured three times in a sitting position after 5 min of rest. The result value of casual systolic and diastolic blood pressure was calculated as the average of the second and third measurement. Patient’s 24-h blood pressure was measured during a hospital stay using an oscillometric device (SpaceLabs 90207, SpaceLabs Medical, Redmond, WA, USA).

### 2.8. Statistical Analysis

Data were processed using the statistical software R 4.0.3 (R Core Team. Foundation for Statistical Computing, Vienna, Austria). Parametric data are presented as an average ± standard deviation. Non-parametric data are presented in the format of median and interquartile range. Study groups and subgroups were compared using the *t*-test for independent samples or the Mann–Whitney U test, depending on the distribution of variables. Categorical data were compared using the Pearson chi-square test or Fisher exact test. *p*-values < 0.05 were considered significant.

Clinically relevant predictors of the PA type which were significant in univariate logistic regression were investigated in the multivariate logistic regression model. Continuous predictors were pre-binarized based on the optimal cut-off value (package cut-off), which was obtained by maximizing of the Youden index (sensitivity + specificity − 1). The final logistics model was selected based on the likelihood ratio test. The SCORE was proposed from the final logistic model, based on the points assigned to individual predictors according to their variable importance (package caret).

The receiver operating characteristic (ROC) analysis included the assessment of optimal cut-off values for SCORE to detect the patients with unilateral or bilateral PA.

## 3. Results

### 3.1. Development Cohort

The diagnosis of PA was confirmed in 263 patients, and 242 patients underwent the AVS, which was successful in 150 of them (failure rate of 38%). Unilateral and bilateral PA was diagnosed in 96 and 54 patients, respectively (Figure 1).

The baseline characteristics of the patients are shown in Table 1. There were no significant differences in age, BMI, and in-office and 24-h blood pressure levels between patients with unilateral and bilateral PA. The proportion of females was significantly higher for unilateral compared with bilateral PA (43% vs. 22%, *p* = 0.01). The patients with unilateral PA had significantly higher baseline PAC (*p* = 0.002), baseline ARR, PAC after the SIT (both *p* < 0.0001), and PAC after the PST (*p* = 0.02). They had significantly lower serum potassium levels (*p* < 0.001) and, as expectated, a significantly higher prevalence of adrenal nodules detected by CT (*p* < 0.0001).

### 3.2. Development of the SCORE

We identified six candidate variables associated with laterality and determined their optimum cut-off values (Table 2). We excluded baseline PAC and PAC after the PST because of collinearity between baseline ARR and baseline PAC and between PAC after the PST and PAC after the SIT, respectively.

The remaining variables were investigated using the univariate (Table 3) and multivariate logistic regression models (Table 4), resulting in three variables, adrenal nodule, PAC after the SIT, and serum potassium concentration, which appeared predictive independently.

The likelihood ratio test showed a good fit in the analysis model (*p* = 0.14). Table 4 also shows points assigned to individual predictors for the construction of the SCORE.

ROC analysis for the ability of the SCORE to diagnose the type of PA is shown in Figure 2, with the area under the ROC curve = 0.834; 95% CI = 0.774–0.893. A histogram of the SCORE is shown in Figure 3. A SCORE = 3 points had a positive predictive value of 100% and sensitivity of 48% for the detection of unilateral PA. On the other hand, a SCORE = 0 had a positive predictive value of 67% and a sensitivity of 61% for the detection of bilateral PA.

### 3.3. Validation Cohort

The diagnosis of PA was confirmed in 263 patients, and 208 patients underwent the AVS, which was successful in 138 of them (failure rate of 34%). Unilateral and bilateral PA was diagnosed in 94 and 44 patients, respectively (Figure 4).

The baseline characteristics of the patients are shown in Table 5. There were no significant differences in proportion of females, age, BMI, and in-office and 24-h blood pressure levels between patients with unilateral and bilateral PA. The patients with unilateral PA had significantly higher baseline serum aldosterone (*p* < 0.0001), baseline ARR (*p* = 0.04), serum aldosterone after the SIT (*p* < 0.0001), and serum aldosterone after the PST (*p* < 0.01). They had significantly lower serum potassium levels previously documented and after the SIT (both *p* < 0.0001) and, by expectation, significantly higher prevalence of adrenal nodules detected by CT (*p* < 0.01).

### 3.4. Validation of the SCORE

ROC analysis for the ability of the SCORE to predict the type of PA is shown in Figure 5, with the area under the ROC curve = 0.729; 95% CI = 0.650–0.809. A histogram of the SCORE is shown in Figure 6. A SCORE = 3 points had a positive predictive value of 100% and sensitivity of 36% for the detection of unilateral PA. On the other hand, a SCORE = 0 had a positive predictive value of 55% and a sensitivity of 55% for the detection of bilateral PA.

## 4. Discussion

We have developed SCORE in order to distinguish unilateral primary aldosteronism based on the presence of adrenal nodules on CT and PAC after the SIT, which are variables which are routinely used in clinical practice and can be readily obtained. Although the global predictive power of SCORE is comparable to several previously published prediction methods, the main finding of the study is that SCORE can identify with 100% specificity and reasonable sensitivity patients with unilateral PA who could hence bypass AVS before surgery. Importantly, there has not been a single patient in both development and validation cohorts with SCORE = 3, who would demonstrate overproduction of aldosterone from the contralateral gland.

Although the AUC ROC does not differ from other scores, our SCORE shows superiority over other previously published scores in the prediction of unilateral PA. Among patients with PA, it can define a group of patients with unilateral PA with 100% certainty, in whom AVS could be omitted before surgery. The largest study on this topic by Kobayashi et al. [24] was focused on the prediction of bilateral PA. From the published data, however, the predictive power of their score ≤ 1 for the detection of unilateral PA can be derived with the positive predictive value of 80% and 85% for the development and validation cohorts, respectively, with respective sensitivities of 42% and 42%. The recently published aldosterone-to-lowest potassium ratio ≥ 15 achieved 91% and 92% positive predictive value for the detection of unilateral PA in the development and validation cohorts, respectively, with respective sensitivities of 46% and 25% [25]. A similar stratification scheme as in our study was published by Kocjan et al. [22]. Their score was able to predict bilateral PA with 100% specificity if the patient did not have a noticeable nodule on CT and had serum potassium above 3.9 mmol/L and PAC after SIT below 180 ng/L. These results, however, remained un-validated. On the other hand, clinically significant prediction of unilateral PA was likely not possible in their cohort, although such results were not reported and cannot be derived from the published data.

Burrello et al. constructed a very sophisticated SPACE score which is able to dichotomize patients at both poles with a high probability into unilateral and bilateral forms of aldosteronism [27]. Our conclusions are not inconsistent with those of Burrello et al. because all the covariates considered by us are also represented in that SPACE score. Maybe due to the smaller number of patients, the strongest predictors of unitary overproduction prevailed among them, i.e., adrenal nodule and PAC value after SIT. Regardless of the presence or absence of nodules on CT of the adrenal glands, we have met hardly any patients who had a PAC value after SIT above 165 ng/L and did not have evidence of unilateral aldosterone overproduction.

Recently, Kocjan et al. have published validation of three novel clinical prediction tools for PA subtyping. In addition to the above-mentioned aldosterone-to-lowest potassium ratio [25] and 20-point clinical prediction SPACE score [27], it was aldosterone concentration and the aldosterone concentration relative reduction rate after SIT [26]. The application of any of the validated clinical prediction tools to their cohort did not predict the PA subtype with the high diagnostic performance originally reported [34]. A major limitation of the published validation was the absence of antihypertensive medication potentially affecting renin-angiotensin-aldosterone system withdrawal. Continued concomitant beta-blockers therapy can lower aldosterone concentration levels and thus theoretically decrease the lateralization index in patients with the lateralized disease. Therefore, some patients with adrenal adenoma and higher aldosterone concentration after SIT may have been misclassified as bilateral overproduction.

The study published by Kobayashi et al. [24] was the only study that respected the generally reported unilateral to bilateral PA ratio (specifically 378 patients with unilateral PA and 912 patients with bilateral PA). A lower percentage of patients with bilateral PA published in other studies (including our study) might lead to a selection bias. Nevertheless, the patients with unilateral PA are more likely to present with resistant hypertension and severe hypokalemia, which leads to the establishment of the diagnosis, and thus, these patients are referred to our department more frequently than those with bilateral PA.

The criterion for the adrenal tumor was ≥6 mm because 6 mm tumor mass is the minimum size of pathological adrenal tissue that can be detected by our institutional multi-slice CT scanner. Higher cut-off values to detect adrenal tumor were used in other studies: Kobayashi et al. [35]: ≥8 mm, Kobayashi et al. [24]: ≥10 mm, Kupers et al. [20]: ≥8 mm, Kamemura et al. [23]: ≥10 mm, and Kocjan et al. [22]: regardless of size. Nevertheless, our criterion of ≥6 mm did not adversely impact the specificity of unilateral PA prediction.

Our study questioned the significe of the role of PST, which was not predictive independently, in classification of unilateral and bilateral PA. The accuracy of PST for the prediction of unilateral PA was slightly lower (51%) compared to previous studies. Mulatero et al. [36] reported an accuracy of 67%. In a smaller study by Lau et al. [37], accuracy for 1-h and 4-h PST was not provided, but the range of 50–60% can be approximated. Espiner et al. [38] found an accuracy of 69% (with 100% specificity), but their cohort of 49 patients included only eight subjects with bilateral PA.

It can be speculated that the reason for the limited predictive power of PST in our population is due to a higher proportion of patients with angiotensin II-responsive at the expense of ACTH-responsive adenomas. Patients with angiotensin II-responsive adenoma have not only baseline PAC comparable to patients with bilateral PA but also respond comparably during PST [39,40,41]. Thus, a higher proportion of such adenomas reduced the power of PST to distinguish between unilateral and bilateral PA and, consequently, the result of the PST was overpowered by SIT in multivariate analysis because post-SIT PAC is less likely dependent on the proportion of adenoma subtypes. On the other hand, the utility of PST in former studies suggests that their populations had a lower proportion of patients with angiotensin II-responsive adenoma [42,43].

The first thorough assessment of SIT in PA patients was conducted in the study by Weigel et al. [44]. Patients PAC > 100 ng/L after the SIT had a higher proportion of unilateral PA compared to the rest of the population (66% vs. 42%; *p* < 0.001). The high predictive value of SIT in distinguishing PA subtypes was recently confirmed in a study by Kaneko et al. [45]. Seated SIT had superior accuracy in subtype diagnosis of PA compared to dexamethasone suppression test.

We are aware of the limitations of our study.

Firstly, the APA group showed a higher proportion of females than the IHA group in the development cohort. As the aldosterone levels may be higher in fertile females than in males, maybe due to the high prevalence of the mutation in the KCNJ5 gene in young females with APA [46,47], a higher proportion of females could contribute to the higher aldosterone level found in the APA group. Nevertheless, the proportion of females was comparable in APA and IHA subgroups in the validation cohort, and the predictive power of the SCORE had not changed substantially.

Secondly, different laboratory tests were used to determine plasma renin and plasma/serum aldosterone levels in the development and validation study. This fact did not allow us to use an internal 10-fold cross validation approach for the validation of SCORE. Although the diagnosis of PA had been changed, the predictive value of SCORE remained acceptable.

Thirdly, the scoring system is likely specific to our center, as there is no universal protocol to diagnose PA across centers with variation in assay methods, cut-off values of screening test, confirmatory testing, and lateralization criteria for AVS. A prospective multicenter study is therefore required to confirm and validate the accuracy of the presented SCORE.

In conclusion, it has been validated within our center that adrenal venous sampling could be omitted before surgery in patients with visible typical Conn´s adenoma on CT scan if they have aldosterone concentration ≥ 165 ng/L after the SIT. This criterion deserves to be investigated in other populations and centers which use a comparable protocol to diagnose PA.

## Figures and Tables

**Figure 1 diagnostics-12-01718-f001:**
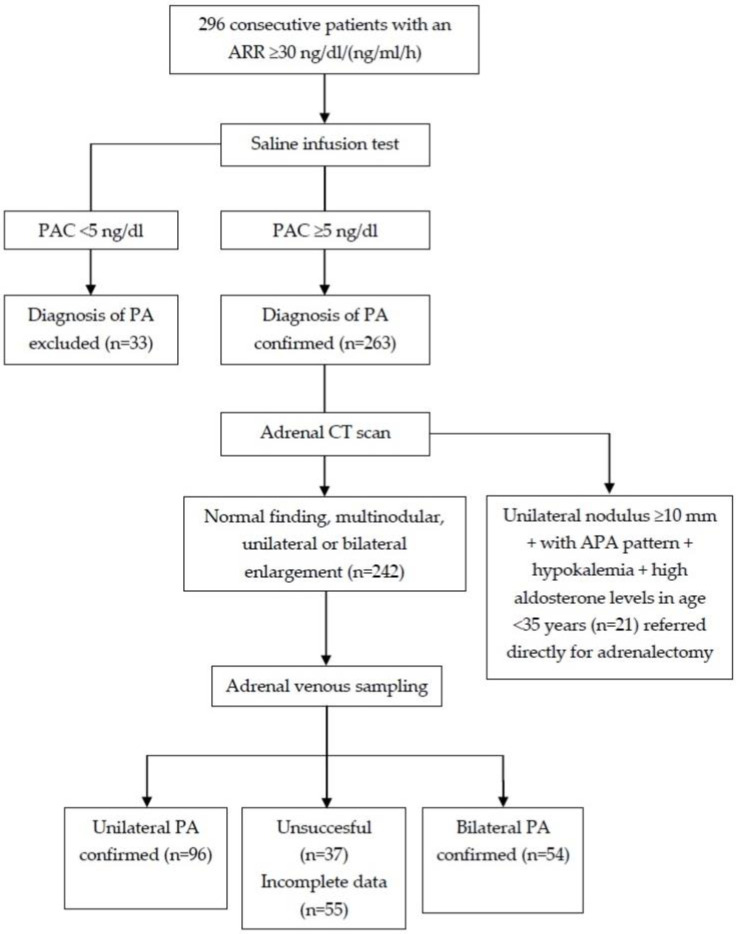
Flow chart of the recruitment into the development cohort. APA—aldosterone producing adenoma; ARR—aldosterone to plasma renin activity ratio; PA—primary aldosteronism; PAC—plasma aldosterone concentration.

**Figure 2 diagnostics-12-01718-f002:**
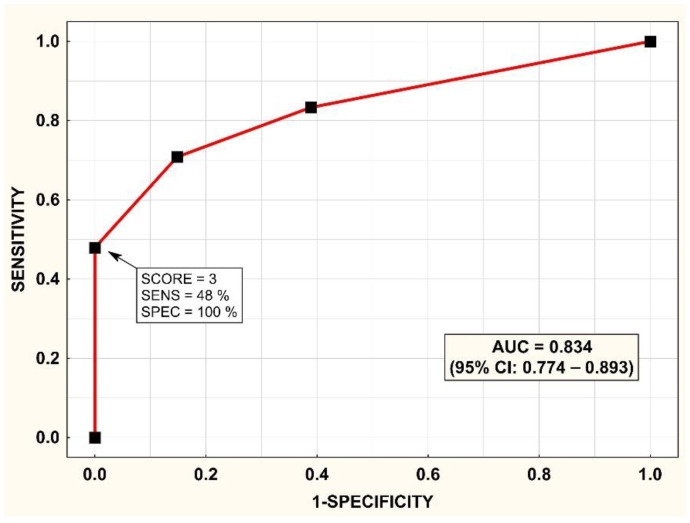
Prediction of unilateral primary aldosteronism (PA) in the development cohort: receiver operating characteristic (ROC) curve for the SCORE. AUC—area under the ROC curve; CI—confidence interval; PA—primary aldosteronism; PPV—positive predictive value; SENS—sensitivity.

**Figure 3 diagnostics-12-01718-f003:**
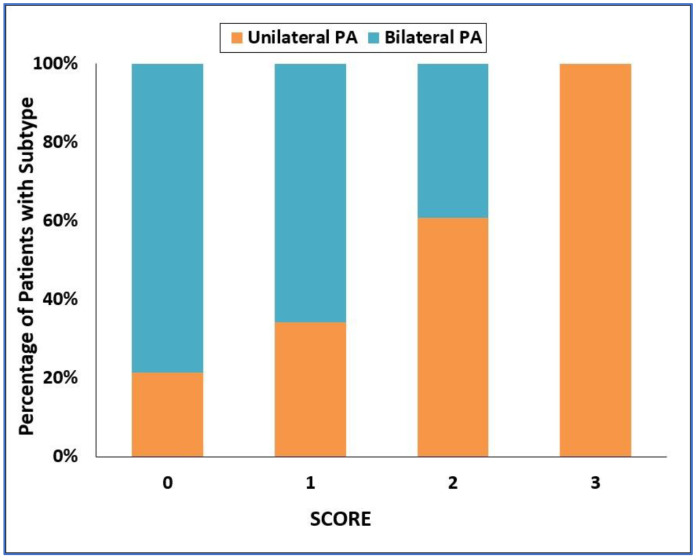
Categorized histogram of the SCORE for unilateral and bilateral primary aldosteronism. PA—primary aldosteronism.

**Figure 4 diagnostics-12-01718-f004:**
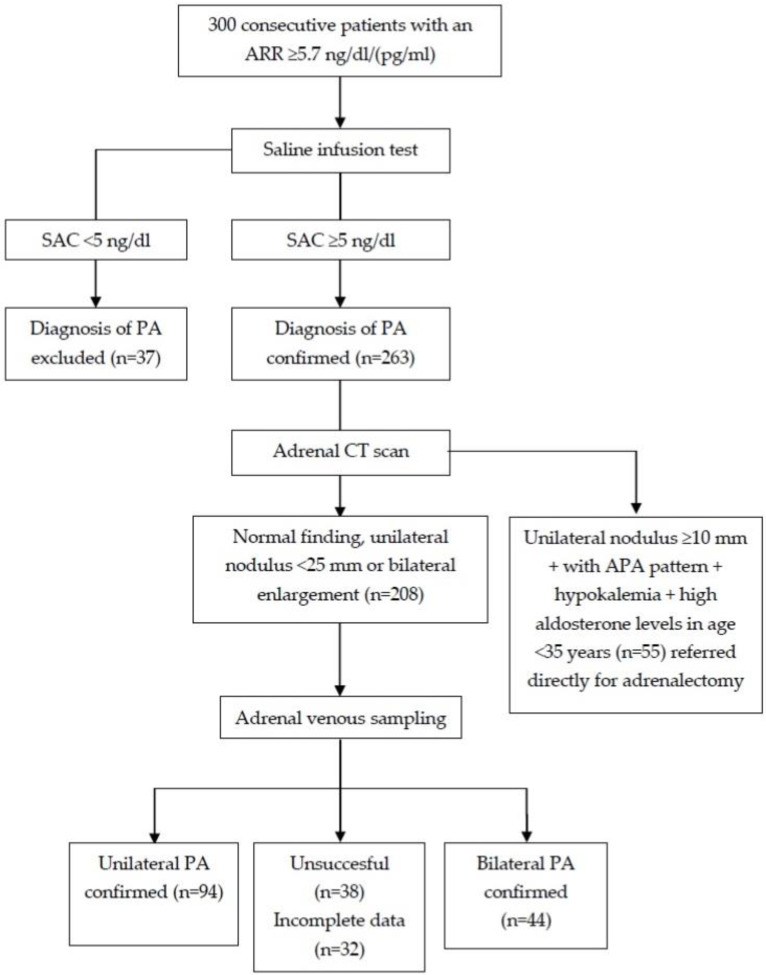
Flow chart of the recruitment into the validation cohort. APA—aldosterone producing adenoma; ARR—aldosterone to plasma renin activity ratio; PA—primary aldosteronism; PAC—plasma aldosterone concentration.

**Figure 5 diagnostics-12-01718-f005:**
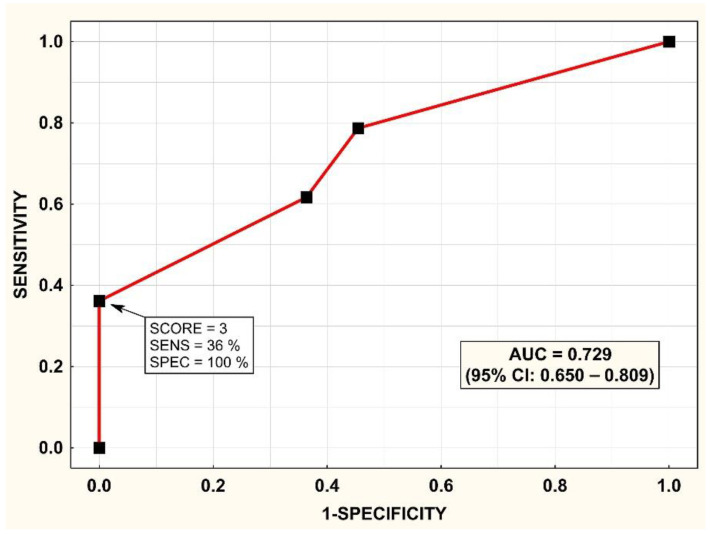
Prediction of unilateral primary aldosteronism (PA) in the validation cohort: receiver operating characteristic (ROC) curve for the SCORE. Abbreviations as in Figure 2.

**Figure 6 diagnostics-12-01718-f006:**
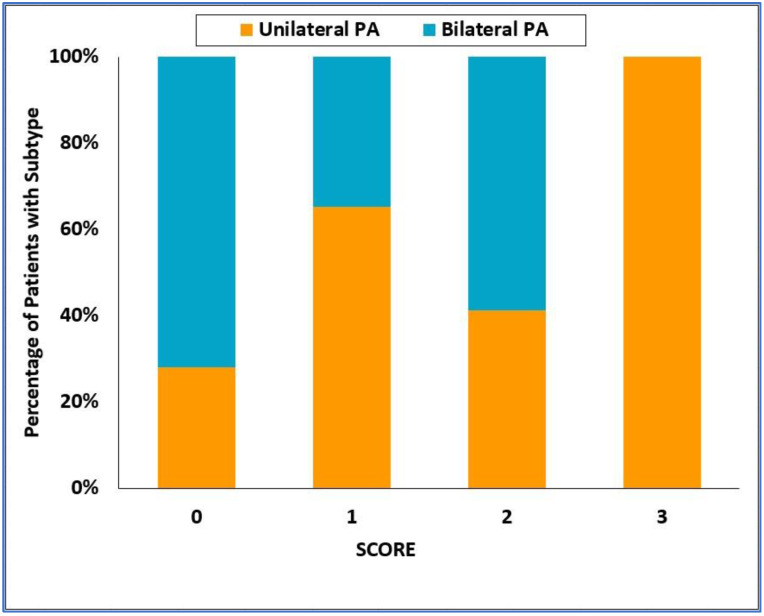
Categorized histogram of the SCORE for unilateral and bilateral primary aldosteronism. PA—primary aldosteronism.

**Table 1 diagnostics-12-01718-t001:** Baseline characteristics of the development cohort.

	Unilateral PA (n = 96)	Bilateral PA (n = 54)	*p*-Value
Age (years)	51 (44–58)	52 (47–58)	0.52
Females	41 (43%)	12 (22%)	0.01
Body mass index (kg/m^2^)	29 (25–32)	30 (28–32)	0.16
Systolic BP (mm Hg)	170 (150–180)	160 (150–170)	0.17
Diastolic BP (mm Hg)	100 (90–110)	97 (88–105)	0.12
24 h systolic BP (mm Hg)	150 (137–161)	150 (137–163)	0.87
24 h diastolic BP (mm Hg)	93 (85–98)	91 (84–97)	0.69
Duration of hypertension (years)	9 (4–15)	11 (6–16)	0.41
Antihypertensive medications (n)	4 (2–5)	4 (2–6)	0.14
Lowest serum potassium recorded (mmol/L)	3.0 (2.8–3.3)	3.2 (2.9–3.5)	0.11
Serum potassium (mmol/L)	3.4 (3.1–3.7)	3.7 (3.4–4.0)	0.0009
Serum potassium < 3.6 mmol/L	58 (60%)	18 (33%)	0.002
eGFR (mL/min/1.73 m^2^)	126 (104–153)	119 (100–133)	0.11
Baseline PAC (ng/L)	291 (183–537)	222 (141–326)	0.002
Baseline PAC ≥ 280 ng/L	49 (51%)	16 (30%)	0.01
Baseline PRA (ng/mL/h)	0.32 (0.20–0.43)	0.35 (0.25–0.53)	0.046
Baseline ARR [ng/dL/(ng/mL/h)]	106 (51–192)	56 (39–91)	<0.0001
Baseline ARR ≥ 100 ng/dL/(ng/mL/h)	51 (53%)	8 (15%)	<0.0001
Prevalence of adrenal nodules on CT	68 (71%)	8 (15%)	<0.0001
PAC after PST (ng/L)	493 (313–821)	378 (310–516)	0.02
Increase in PAC after PST < 30%	30 (31%)	8 (15%)	0.03
PAC after SIT (ng/L)	180 (121–348)	115 (81–163)	<0.0001
PAC after SIT ≥ 165 ng/L	58 (60%)	13 (24%)	<0.0001

Values are shown as medians (interquartile range) or absolute values (percentages). ARR—aldosterone-to-renin ratio; BP—blood pressure; eGFR—estimated glomerular filtration rate = 194 × (SCr)^−1.094^ × (Age)^−0.287^; ×0.739 if female; CT—computed tomography; PA—primary aldosteronism; PAC—plasma aldosterone concentration; PRA—plasma renin activity; PST—postural stimulation test; SIT—saline infusion test.

**Table 2 diagnostics-12-01718-t002:** Diagnostic values of the screening tests for unilateral PA in the development cohort.

	AUC ROC (95% CI)	Cut-Off Value	SENS	SPEC	PPV	NPV	ACC	*p*-Value
Adrenal nodule on CT	0.780 (0.714–0.846)	≥6 mm	71%	85%	90%	62%	76%	<0.001
Serum potassium	0.635 (0.555–0.716)	<3.6 mmol/L	60%	67%	76%	49%	63%	0.005
ARR baseline	0.692 (0.616–0.768)	≥100 ng/dL/(ng/mL/h)	53%	85%	86%	51%	65%	<0.001
PAC baseline	0.648 (0.569–0.727)	≥280 ng/L	51%	70%	75%	45%	56%	0.01
PAC increase after PST	0.582 (0.501–0.663)	<30%	31%	85%	79%	41%	51%	0.05
PAC after SIT	0.682 (0.606–0.757)	≥165 ng/L	60%	76%	82%	52%	66%	<0.001

ACC—accuracy; ARR—aldosterone-to-renin ratio; AUC-ROC—area under the receiver operating characteristic curve; CI—confidence interval; CT—computed tomography; NPV—negative predictive value; PAC—plasma aldosterone concentration; PPV—positive predictive value; PST—postural stimulation test; SENS—sensitivity; SIT—saline infusion test; SPEC—specificity.

**Table 3 diagnostics-12-01718-t003:** Candidate factors for the prediction of unilateral PA (univariable and multivariable logistic regression analysis).

	Univariable Analysis		Multivariable Analysis	
Factor	OR (95% CI)	*p*-Value	OR (95% CI)	*p*-Value
Adrenal nodule	14.0 (5.9–33.3)	<0.001	10.9 (4.3–27.4)	<0.001
Serum potassium < 3.6 mmol/L	2.9 (1.4–5.6)	0.003	1.9 (0.8–4.5)	0.13
ARR baseline ≥ 100 ng/dL/(ng/mL/h)	2.4 (0.6–9.1)	0.22	2.8 (0.4–8.4)	0.49
PAC after SIT ≥ 165 ng/L	4.8 (2.3–10.1)	<0.001	3.1 (1.3–7.4)	0.01
Female sex	3.0 (1.4–6.6)	0.005	1.4 (0.5–3.7)	0.50

CI—confidence interval; OR—odds ratio; PAC—plasma aldosterone concentration; SIT—saline infusion test.

**Table 4 diagnostics-12-01718-t004:** Significant factors for the prediction of unilateral PA.

Factor	OR (95% CI)	*p*-Value	β Coefficient	Points
Adrenal nodule	10.9 (4.3–27.4)	<0.001	2.38	2
PAC after SIT ≥ 165 ng/L	3.1 (1.3–7.4)	0.01	1.11	1

CI—confidence interval; OR—odds ratio; PAC—plasma aldosterone concentration; SIT—saline infusion test. Points—assigned points for the calculation of the SCORE.

**Table 5 diagnostics-12-01718-t005:** Baseline characteristics of the validation cohort.

	Unilateral PA (n = 94)	Bilateral PA (n = 44)	*p*-Value
Age (years)	49 (41–57)	47 (41–56)	0.55
Females	23 (24%)	13 (30%)	0.53
Body mass index (kg/m^2^)	30 (27–33)	31 (29–34)	0.16
Systolic BP (mm Hg)	159 (150–170)	155 (145–162)	0.12
Diastolic BP (mm Hg)	99 (90–105)	98 (90–103)	0.68
24 h systolic BP (mm Hg)	150 (140–158)	147 (139–157)	0.48
24 h diastolic BP (mm Hg)	91 (85–96)	91 (85–98)	0.96
Duration of hypertension (years)	8 (5–12)	7 (3–16)	0.78
Antihypertensive medications (n)	4 (2–4)	3 (2–5)	0.67
Lowest serum potassium recorded (mmol/L)	3.2 (2.9–3.5)	3.6 (3.3–3.9)	<0.0001
Serum potassium (mmol/L)	3.4 (3.2–3.7)	3.9 (3.7–4.1)	<0.0001
Serum potassium < 3.6 mmol/L	56 (60%)	8 (18%)	<0.0001
eGFR (ml/min/1.73 m^2^)	128 (114–165)	152 (110–186)	0.24
Baseline serum aldosterone (ng/L)	265 (187–365)	168 (131–208)	<0.0001
Baseline serum aldosterone ≥ 280 ng/L	41 (46%)	2 (5%)	<0.0001
Baseline DRC (ng/mL)	1.50 (0.49–2.70)	1.51 (0.50–3.30)	0.62
Baseline ARR [ng/dL/(ng/mL)]	13 (6–35)	9 (6–15)	0.04
Prevalence of adrenal nodule on CT	58 (62%)	16 (36%)	0.005
Serum aldosterone after PST (ng/L)	316 (231–446)	233 (192–361)	0.008
Increase in serum aldosterone after PST < 30%	54 (61%)	11 (27%)	0.0003
Serum aldosterone after SIT (ng/L)	175 (117–232)	110 (75–139)	<0.0001
Serum aldosterone after SIT ≥165 ng/L	50 (53%)	4 (9%)	<0.0001

Values are shown as medians (interquartile range) or absolute values (percentages). ARR—aldosterone-to-renin ratio; BP—blood pressure; DRC—direct renin concentration; eGFR—estimated glomerular filtration rate = 194 × (SCr)^−1.094^ × (Age)^−0.287^; ×0.739 if female; CT—computed tomography; PA—primary aldosteronism; PST—postural stimulation test; SIT—saline infusion test.

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
