# Peer review of "Adrenal Venous Sampling Could Be Omitted before Surgery in Patients with Conn’s Adenoma Confirmed by Computed Tomography and Higher Normal Aldosterone Concentration after Saline Infusion Test"

_diagnostics, 2022, doi:10.3390/diagnostics12071718_

Round 1

Reviewer 1 Report

Holaj et al. developed herein a SCORE to predict unilateral Conn's adenoma, without performing invasive adrenal vein sampling, taking into account imaging, serum potassium and aldosterone levels after SIT. Although the study is well designed and its results clearly described, it lacks in originality, as several previous attempts with similar conclusions exist in the literature.

 My major concerns:

Hypokalemia upon supplementation, as already discussed by the authors, cannot provide an objective assessment of its extent. Inclusion as such in the SCORE is risky. Instead, the presence (defined either by supplementation or by K+ <3.5 mmol/l) or absence could be more objective in the SCORE.

 Considering that AVS is an invasive diagnostic method, it would be much more relevant to avoid it in cases of bilateral disease where medical treatment is the only option. In this context, a lower cut-off for the diagnosis of bilateral disease would be clinically more useful. Could the authors make such a suggestion as well?

 Application of other scores suggested in the literature (eg. SPACE or Aldo relative reduction after SIT) in the validation cohort could give us hints about the superiority of the present SCORE.

 Minor points:

-Biochemical parameters mentioned in the methods section are not presented in the results (Urea, Creatinine, total cholesterol, HDL, LDL, triglycerides and glucose).

-eGFR definition is not described in the methods

-SIT methodology: measurement of cortisol to ensure ACTH independency?

-line 288: SAT replace with SIT

Author Response

Holaj et al. developed herein a SCORE to predict unilateral Conn's adenoma, without performing invasive adrenal vein sampling, taking into account imaging, serum potassium and aldosterone levels after SIT. Although the study is well designed and its results clearly described, it lacks in originality, as several previous attempts with similar conclusions exist in the literature.

My major concerns:

  1. Hypokalemia upon supplementation, as already discussed by the authors, cannot provide an objective assessment of its extent. Inclusion as such in the SCORE is risky. Instead, the presence (defined either by supplementation or by K+ <3.5 mmol/l) or absence could be more objective in the SCORE.

Thank you for your valuable comments.

Based on the second reviewer's comment, the baseline ARR value was used as a covariate in the univariate and multivariate analysis instead of the baseline PAC value in the new version of the manuscript. However, ARR did not enter the prediction model as a statistically significant covariate, nor did gender and, surprisingly, neither did hypokalemia. The resulting SCORE now only includes the presence of an adrenal nodule and PAC after SIT.

  1. Considering that AVS is an invasive diagnostic method, it would be much more relevant to avoid it in cases of bilateral disease where medical treatment is the only option. In this context, a lower cut-off for the diagnosis of bilateral disease would be clinically more useful. Could the authors make such a suggestion as well?

We fully agree with this opinion. In our work, as well as in the works of many other authors (Kobayashi et al. J Hypertens 2018 and Puar et al. J Hypertens 2020 etc.), no predictive model succeeded in achieving a 100% predictive value of the bilateral form of the disease. This is probably related to the greater variability of the laboratory values of the found covariates forming individual prediction scores in patients with a bilateral form of the disease.

  1. Application of other scores suggested in the literature (eg. SPACE or Aldo relative reduction after SIT) in the validation cohort could give us hints about the superiority of the present SCORE.

Comparison of our proposed SCORE with others published in the literature including SPACE (Burrello et al. JCEM 2021) will be the subject of follow-up work that we are currently finalizing.

Minor points:

  1. Biochemical parameters mentioned in the methods section are not presented in the results (Urea, Creatinine, total cholesterol, HDL, LDL, triglycerides and glucose).

Biochemical parameters not presented in the results were omitted in the methods section in the new version of the manuscript.

  1. eGFR definition is not described in the methods

estimated glomerular filtration rate (eGFR) = 194 x (SCr)-1.094 x (Age)-0.287; x 0.739 if female;

eGFR definition was newly described in the methods in the new version of the manuscript.

  1. SIT methodology: measurement of cortisol to ensure ACTH independency?

All patients admitted to our department in order to rule out secondary hypertension undergo a 1 mg dexamethasone suppression test to rule out cortisol hypersecretion. Plasma cortisol during the test has been suppressed in all patients. Plasma cortisol in our patients is also determined during the postural stimulation test and, of course, also during the AVS.

  1. line 288: SAT replace with SIT

SAT was replaced with SIT in the new version of the manuscript.

Reviewer 2 Report

Holaj R et al. aimed to identify non-invasive characteristics that can select patients with unilateral Primary Aldosteronism (uPA) who could hence bypass AVS before surgery. They proposed a score (called SCORE), which was calculated as a sum of the prevalence of adrenal nodules ≥6 mm on computed tomography (5 points), plasma/serum aldosterone concentration ≥165 ng/l after the saline infusion test (SIT) (3 points), and serum potassium <3.6 mmol/l (2 points). The SCORE ≥8 points identified 48% of uPA patients with a specificity of 100% in the development cohort. With the same cut-off value in the validation cohort, the zero rate of false-positive classifications was preserved. They argued that AVS could be omitted before surgery in patients with typical Conn´s adenoma if they have aldosterone concentration ≥165 ng/l after the SIT.

The topic is interesting and debate. Also the paper is potentially interesting, however there are several major concerns to be solved before the paper could be acceptable for publication. The major unsolved problems are:

- the lack of an external validation cohort, as stated by the authors in the discussion;

- the presence of an internal validation cohort, diagnosed using different laboratory methods.

Majors
- Creating a clinical score dichotomizing the considered variables is a simplification of a complex and interesting model. This is a statistical pity, which could be acceptable for clinical utility, but it remains a problem when the single and global accuracy of the variables and the model are moderate and the final outcome is the diagnosis of a surgically correctable form of PA. Therefore, the derived conclusions should be treated with extreme caution. In any case, I think that only an elegant model, treating variables ad continuous, instead of a score, should be more correct in this specific case, even if not friendly and easy to use as the SCORE.

- The selection of the variables for the logistic regression model has to be recontrolled. Why did the authors exclude baseline ARR? In my opinion a better evaluation should be done considering the predictive power of adrenal nodules, serum potassium, baseline ARR and PAC after SIT.

- Moreover, points should be attributed on the basis of the beta-coefficients (that are lacking among the paper and should be added) and the covariate-assigned points are questionable. In fact, PAC after SIT has a Beta-coefficient of 1.16, while serum potassium of 0.91; in my opinion there is not a real difference for attributing different points to these two covariates.

- If 6 mm is the minimum detectable size by the CT of the authors’ center, I think it could be more correct to consider this variable as a unilateral adrenal nodule and not as adrenal nodule over 6 mm.

- Therefore, all the considerations are overstated. Accuracies should be reassessed for all the models.

- How many patients did undergo 1 mg dexamethasone suppression test? It is not clear how cortisol co-secretion has been excluded in the enrolled population. This point is extremely important for assessing lateralization during AVS, but also for considering the impact of aldosterone levels in diagnosis and in the SCORE.

Minors
- Adrenal nodulus should be replaced by adrenal nodule among the paper.

- Authors should specify precise criteria that led them considering patients with lateralization index between 2 and 4 affected by uPA.

- It is strange that female patients have a significant higher rate of uPA, compared to male gender, but in the univariate logistic regression male sex is associated with uPA. The association is not significant, but I would like to be sure that the variable considered was really the male gender and not the female one.

- I suggest adding these references regarding resistant hypertension and PA and after the statement on line 300-301 (10.1097/HJH.0000000000002441, 10.7326/M20-0065)

- References should be edited because numbers in text are not always rightly associated with articles in the reference list.

Author Response

Holaj R et al. aimed to identify non-invasive characteristics that can select patients with unilateral Primary Aldosteronism (uPA) who could hence bypass AVS before surgery. They proposed a score (called SCORE), which was calculated as a sum of the prevalence of adrenal nodules ≥6 mm on computed tomography (5 points), plasma/serum aldosterone concentration ≥165 ng/l after the saline infusion test (SIT) (3 points), and serum potassium <3.6 mmol/l (2 points). The SCORE ≥8 points identified 48% of uPA patients with a specificity of 100% in the development cohort. With the same cut-off value in the validation cohort, the zero rate of false-positive classifications was preserved. They argued that AVS could be omitted before surgery in patients with typical Conn´s adenoma if they have aldosterone concentration ≥165 ng/l after the SIT.

The topic is interesting and debate. Also the paper is potentially interesting, however there are several major concerns to be solved before the paper could be acceptable for publication. The major unsolved problems are:

- the lack of an external validation cohort, as stated by the authors in the discussion;

- the presence of an internal validation cohort, diagnosed using different laboratory methods.

Majors
1. Creating a clinical score dichotomizing the considered variables is a simplification of a complex and interesting model. This is a statistical pity, which could be acceptable for clinical utility, but it remains a problem when the single and global accuracy of the variables and the model are moderate and the final outcome is the diagnosis of a surgically correctable form of PA. Therefore, the derived conclusions should be treated with extreme caution. In any case, I think that only an elegant model, treating variables ad continuous, instead of a score, should be more correct in this specific case, even if not friendly and easy to use as the SCORE.

Thank you for your valuable comments.

I fully agree with this opinion. I greatly appreciate the work of the authors Burrello et al. JCEM 2021, who constructed a very sophisticated and statistically hard-to-dispute SPACE score that is able to dichotomize patients at both poles with high probability into unilateral and bilateral forms of aldosteronism. In the conditions of our single center, it is very difficult to compete in any way with such multicenter collaboration. On the contrary, we tried to implement our own findings from the course of over twenty years to our work. Our conclusions are not inconsistent with those of Burrello et al. All the covariates considered by us are also represented in the work of Burrello et al. Maybe due to the smaller number of patients, the strongest predictors of unitary overproduction prevailed among them, i.e. adrenal nodule and PAC value after SIT. Regardless of the presence or absence of nodule on CT of the adrenal glands, we have not met a patient in our entire twenty-year career who had a PAC value after SIT above 165 nmol/l and did not have evidence of unilateral aldosterone overproduction. That is why our conclusions may seem so bold. However, this is only a small group of our patients. For all others, performing an AVS before deciding on the correct treatment strategy is still indicated.

  1. The selection of the variables for the logistic regression model has to be recontrolled. Why did the authors exclude baseline ARR? In my opinion a better evaluation should be done considering the predictive power of adrenal nodules, serum potassium, baseline ARR and PAC after SIT.

Default ARR was used in the new version of the manuscript instead of the PAC as a covariate in both univariate and multivariate analysis. However, ARR did not enter the prediction model as a statistically significant covariate, nor did gender, and surprisingly, neither did potassium value. In the new version, the resulting SCORE only includes the presence of an adrenal nodule and the PAC value after SIT.

  1. Moreover, points should be attributed on the basis of the beta-coefficients (that are lacking among the paper and should be added) and the covariate-assigned points are questionable. In fact, PAC after SIT has a Beta-coefficient of 1.16, while serum potassium of 0.91; in my opinion there is not a real difference for attributing different points to these two covariates.

Points were now assigned based on beta-coefficients, namely 2.38 for adrenal nodule and 1.11 for PAC value after SIT.

  1. If 6 mm is the minimum detectable size by the CT of the authors’ center, I think it could be more correct to consider this variable as a unilateral adrenal nodule and not as adrenal nodule over 6 mm.

The size of the nodule characterizing the variable is being omitted in the new version of the manuscript.

  1. Therefore, all the considerations are overstated. Accuracies should be reassessed for all the models.

Whether we used the baseline value of PAC or the baseline value of ARR as a variable in the multivariate analysis. Only the presence of a nodule and PAC after SIT were identified as significant predictors in the new version of SCORE. Hypokalemia did not reach statistical significance as a predictor in the multivariate analysis and therefore did not enter the prediction model.

  1. How many patients did undergo 1 mg dexamethasone suppression test? It is not clear how cortisol co-secretion has been excluded in the enrolled population. This point is extremely important for assessing lateralization during AVS, but also for considering the impact of aldosterone levels in diagnosis and in the SCORE.

All patients admitted to our department in order to rule out secondary hypertension undergo a 1 mg dexamethasone suppression test to rule out cortisol hypersecretion. Plasma cortisol during the test has been suppressed in all patients. Plasma cortisol in our patients is also determined during the postural stimulation test and, of course, also during the AVS.

Minors
7. Adrenal nodulus should be replaced by adrenal nodule among the paper.

Adrenal nodulus was replaced by adrenal nodule in the new version of the manuscript.

  1. Authors should specify precise criteria that led them considering patients with lateralization index between 2 and 4 affected by uPA.

Criteria for consideration of patients with lateralization index (LI) between 2 and 4 as inflicted by unilateral aldosteronism were determined according to a Position statement and consensus of the Working Group on Endocrine Hypertension of the European Society of Hypertension as:

"AVS in which lateralization index are between 2 and 4 could be candidates for surgery in some cases when other parameters and clinical data are taken into account, such as the presence of contralateral suppression, lack of response to medical therapy, side effects, and so forth" (Mulatero et al. J Hypertens 2020).

We used contralateral ratio <1 (what may indicate unilateral aldosteronism on the opposite side), hypokalemia below 3,0 mmol/l, and persistence of uncontrolled hypertension on MRA therapy.

Precise criteria leading us considering patients with lateralization index between 2 and 4 inflicted by uPA were specified in the new version of the manuscript.

  1. It is strange that female patients have a significant higher rate of uPA, compared to male gender, but in the univariate logistic regression male sex is associated with uPA. The association is not significant, but I would like to be sure that the variable considered was really the male gender and not the female one.

It is  true. The variable under consideration should have indeed been female gender.

  1. I suggest adding these references regarding resistant hypertension and PA and after the statement on line 300-301 (10.1097/HJH.0000000000002441, 10.7326/M20-0065).

These two references have been added in the new version of the manuscript.

  1. References should be edited because numbers in text are not always rightly associated with articles in the reference list.

All references have been checked and inappropriate edited in the new version of the manuscript.

Round 2

Reviewer 1 Report

The authors adequately addressed my comments. No further comments from my side. 

Reviewer 2 Report

Authors solved all the previously rised concerns. In my opinion he article is suitable for publication.